# Protein Interaction with Charged Macromolecules: From Model Polymers to Unfolded Proteins and Post-Translational Modifications

**DOI:** 10.3390/ijms20051252

**Published:** 2019-03-12

**Authors:** Pavel Semenyuk, Vladimir Muronetz

**Affiliations:** 1Belozersky Institute of Physico-Chemical Biology, Lomonosov Moscow State University, 119234 Moscow, Russia; vimuronets@belozersky.msu.ru; 2Faculty of Bioengineering and Bioinformatics, Lomonosov Moscow State University, 119234 Moscow, Russia

**Keywords:** electrostatic interactions, molecular dynamics simulations, post-translational modification, polyelectrolyte, protein‒polyelectrolyte complex, sulfation, glycation

## Abstract

Interaction of proteins with charged macromolecules is involved in many processes in cells. Firstly, there are many naturally occurred charged polymers such as DNA and RNA, polyphosphates, sulfated glycosaminoglycans, etc., as well as pronouncedly charged proteins such as histones or actin. Electrostatic interactions are also important for “generic” proteins, which are not generally considered as polyanions or polycations. Finally, protein behavior can be altered due to post-translational modifications such as phosphorylation, sulfation, and glycation, which change a local charge of the protein region. Herein we review molecular modeling for the investigation of such interactions, from model polyanions and polycations to unfolded proteins. We will show that electrostatic interactions are ubiquitous, and molecular dynamics simulations provide an outstanding opportunity to look inside binding and reveal the contribution of electrostatic interactions. Since a molecular dynamics simulation is only a model, we will comprehensively consider its relationship with the experimental data.

## 1. Introduction

The importance of electrostatic interaction for biological processes is clear since cells contain a lot of charged molecules (Figure 1). First, literally all proteins are polyelectrolytes since they contain charged amino acids, i.e., aspartate, glutamate, lysine, arginine, and histidine [1]. Depending on the pH of the system, proteins are charged either positively or negatively, and many proteins such as actin, tubulin, serum albumin, histones, lysozyme, cytochrome c, etc., are strongly charged under physiological conditions [2]. Even if a net charge of the protein molecule is small, it usually has negatively and positively charged areas on the surface, which are important for the interaction of the protein with other macromolecules. A significant change in local charge can alter protein behavior. From this point of view, post-translational modifications such as phosphorylation and polyphosphorylation [3], sulfation [4,5], glycation [6,7], oxidation [8], and polysialylation [9] are of special interest since they can strongly influence protein‒protein interactions.

The second class of naturally occurred charged macromolecules is comprised of nucleic acids. Electrostatic interactions are important for the interaction of DNA with histones and chromatin condensation [10], interactions of ribosomal proteins with ribosomal RNA [11], as well as for the formation of other protein‒DNA and protein‒RNA complexes [12,13]. Moreover, there are many charged polysaccharides—sulfated or carboxylated glycosaminoglycans such as heparin, heparan sulfate, hyaluronic acid, etc.—which form the extracellular matrix, often being a component of proteoglycans [14]. The list of similarly charged polysaccharides from plants includes sulfated polymers such as carrageenan and fucoidan, carboxylated polymers such as alginic acid, pectin, etc. Some of them are widely used in food chemistry [15,16] or suggested for medicinal use [17,18]. Finally, we must mention phosphate-containing macromolecules, which are widely present in live cells: from ubiquitous molecules such as ATP and inositol trisphosphate to various linear polyphosphates that are synthesized in prokaryotes and eukaryotes [19,20] and can be attached to proteins [3,21].

Many of the aforementioned systems are extremely difficult to study in vivo and even in vitro. From this point of view, molecular modeling provides an outstanding opportunity to look at the protein interaction, with numerous macromolecules moving to the atomistic level [22,23]. Classical molecular dynamics (MD) simulations, which deal with all-atom structures and are limited by Newton’s laws of motion, allow for studying molecular movements and interactions in terms of dynamics. In addition, classical MD approach can be expanded to both directions: downstairs, i.e., a more detailed view, using combined quantum mechanics/molecular mechanics (QM/MM) approach [24]; and upstairs, i.e., a rougher investigation, using coarse-grained MD simulations [25,26]. In the present mini-review, we consider the interaction of proteins with the aforementioned charged macromolecules, focusing on the modeling of the electrostatic interactions. However, since realistic modeling of charged molecules behavior is a challenge, we comprehensively compare the data from molecular dynamics simulations with the experimental data throughout the review.

## 2. Protein Interaction with Model Polymers and Nucleic Acids

Homopolymers and simple (from a biological point of view) copolymers of charged repeat units, mainly synthetic polymers, can be considered as a simplified model to study electrostatic interactions in experiments as well as in modeling according to the assumption that such interactions are generally charge-driven. Computer modeling of such model systems can be performed on a different detailing level, i.e., using different approaches: Monte Carlo [27,28,29,30,31], Brownian [32], or Langevin dynamics simulations [33,34], as well as coarse-grained [35] and atomistic molecular dynamics simulations [36,37]. The use of these computational approaches is discussed in numerous reviews such as [38,39,40].

Several groups adopted molecular dynamics simulations approach to investigate protein interaction with charged polysaccharides because of high biological impact of polysaccharides as important components of extracellular matrix [41]. Thus, atomistic MD simulations were used for investigation of the interaction of heparin, heparan sulfate, and other glycosaminoglycans with interleukin 8 [42,43], cell growth factors [44], sclerostin [45], chemokine CCL5 [46], bone morphogenetic protein 2 [47], and Aβ_1-42_ fibrils [48]. In addition, heparin and its analogues are of special interest as anticoagulation agents [49,50], and therefore MD simulations of their interaction with various coagulation factors [51] are necessary for new anticoagulation drug design. Technically, MD simulations of such systems are usually relatively simple since a wide range of monomers (including sulfated ones) are already well parameterized in the GLYCAM force field, which contains parameters for numerous carbohydrates [52]. For short polymers such as low-molecular-weight heparin or heparan sulfate oligomers, an initial hypothesis on the binding site location can be made using docking [44,45] or sometimes using the crystal structure of the protein complex with a similar polymer [53]. In addition to accurate refinement of the binding site, the MD simulations approach is useful to compare different glycasaminoglycans since this is a very diverse class of polymers. Thus, such a comparison study, combined with experimental data, showed that the sulfation of glycasaminoglycans enhances the binding with protein, and furthermore, not only the number of sulfate groups but also the sulfation pattern is important for the binding specificity [43,45,46].

Computational studies of the protein interaction with polyphosphates, another class of natural polyanions, are less numerous (apart from ATP, ADP, etc., which are sometimes called polyphosphates). However, MD simulations were adopted to investigate atomistic details of specific [54] and non-specific interactions [37,55]. In all cases, the binding was shown to be electrostatically driven. The main conclusions (but not detailed interaction mechanism on the atomistic level) were indirectly corroborated with the experiments.

Next, we should mention modeling of the cationic or anionic dendrimers, which are extensively studied as multi-functional drug carriers [56,57] in complexes with different proteins: immunoglobulins [58], amyloidogenic prion protein [59], actin [60], human serum albumin [61], HIV-derived peptides [62,63], ferritin [64], and other proteins [65]. Interestingly, control of protonation state (and therefore the net charge) of polyamidoamine (PAMAM) dendrimers allowed control of their interaction with cytolysin A, a protein toxin forming pores in cell membrane and thus causing membrane permeabilization [66]. The binding resulted in pore closure and inhibition of cytolysin toxic activity, nicely illustrating the importance of electrostatic interactions.

Comprehensive investigations of the protein complexes with charged model polymers revealed the regularities and key factors of these interactions. The first lesson learned from the study of such model systems is a deep understanding of patch-based interaction of similarly charged macromolecules. Indeed, proteins have both negatively and positively charged patches on the surface in a wide pH range, which can interact with oppositely charged regions of the polymers even if total charges of the protein and the polymer are the same [39,59,67]. The patch-based interaction model helped us to explain the chaperone-like activity of synthetic polyelectrolytes [68,69,70,71], which can be higher in the case of similarly charged proteins and polyelectrolytes [67]. Using MD simulations, we suggested that long polyelectrolyte chains bind the protein via only a part of the monomers, and the unbound monomers form charged loops and tails around the protein surface, providing stability for the complex. Thus, protective efficiency increased in the case of similarly charged proteins and polymers since the oppositely charged patches on the protein surface are smaller (Figure 2). The suggested model was corroborated by experiments [37,67].

Some technical caution is required to obtain a realistic model of protein interaction with highly charged polymers. In principle, electrostatic interactions are handled as non-bonded interactions of charged atoms (or atom groups). Long-range electrostatic interactions can be expressed using the particle mesh Ewald (PME) technique [72], which is a gold standard for highly charged molecules such as nucleic acids [73,74] and sulfated polysaccharides. Obviously, accurate parametrization of charged monomers (residues), especially atom partial charges values, is of special importance for realistic modeling of the aforementioned systems. There are many popular force fields designed for nucleic acids; the GLYCAM force field provides parametrization of numerous polysaccharide monomers including sulfated glycosaminoglycans [52]. However, the choice of the force field for mixed systems can be difficult. For some protein post-translational modification products (reviewed below), the parameters for GROMOS and AMBER force fields were developed [75,76,77]. For other non-standard monomers and residue modifications, accurate determination of partial atom charges using RED III tools [78], PRODRG server [79], ATB tools [80] or other tools is required.

According to experimental data and theoretical calculations, one of the key factors in the formation of inter-polyelectrolyte complexes is counterion release [81,82,83,84,85], in addition to other factors [86,87,88,89]. In the case of polymers with high charge density, low-molecular-weight counterions are partially bound with a charged chain, but the binding with an oppositely charged polymer results in the release of counterions bound with both polymers, associated with entropy gain [40,81,90]. This effect was obtained in atomistic MD simulation of the polymers with counterions [91]. Unfortunately, its realistic modeling in complex multicomponent systems is difficult since popular small molecular ions models were shown to overestimate Coulomb interactions (the same might be true for all ionic interactions [92]), which results in an artificial crystallization in the solution of ions (probably because of the polymer molecule overcharging, similar to the charge inversion observed in a polyelectrolyte solution with multivalent counterions [93]). Enhanced ion models by Joung and Cheatham seem to solve this problem in the case of pure salt solutions and probably for “generic” biomolecules [94]. Unfortunately, even this model does not provide a realistic simulation in the case of molecules with high charge density. Indeed, testing of both ion models (i.e., standard one from AMBER99-parmbsc0 force field and Joung‒Cheatham model) for poly(methacrylic acid) gave the same artificial aggregation of the polyelectrolyte in solution with counterions [95]. The same was true for the simulation of polyphosphate using two different ion models, the standard from GROMOS 54a7 force field and the Joung‒Cheatham model [37], but not for DNA, poly(styrene sulfonate), and pyridinium polycations, in which the distance between charged groups is higher and therefore the charge density is lower [37,67,95].

However, there are many good examples of realistic modeling of the inter-polyelectrolyte interaction accompanied by counterion release or binding, and even attempts to compare different counterions [96]. As an example, we can mention simulations of DNA complexation and decomplexation with polycations in a salt solution [97,98]. Summarizing the above, we conclude that, despite some problems with realistic parametrization of low-molecular weight ions requiring some care, the overall behavior of common systems can be simulated realistically.

Nucleic acids belong to one more important class of polyanions present in a cell. Obviously, the binding of proteins with nucleic acids is determined by many of the following factors, but electrostatic interactions are important for the binding [99]. Generally, electrostatic interactions provide non-specific binding with backbone phosphate groups, but sequence-specific binding can also partly arise from electrostatic interactions [12] since AT-rich and GC-rich sequences display the difference in electrostatic potential value and polarity of minor and major grooves [100]. As for protein structure, positive charge distribution can be important for DNA-binding proteins, not only the value of the charge [101]. In addition, electrostatic interactions indirectly influence protein‒DNA binding, associated with DNA bending or kinking [12], since they partially determine double-stranded DNA rigidity [102]. Therefore, careful handling of electrostatic interactions is of special importance for modeling the protein interaction with DNA and RNA. There are two ways to obtain the complex: MD simulation of individual protein and the addition of nucleic acid molecules until the binding is achieved (this way may require some sampling), and the improvement of the docking results using MD simulation, which is of special importance for protein complexes with flexible RNA molecules [99]. Because of numerous reviews on the MD simulations of RNA and DNA protein complexes, we do not focus on them here and only suggest a few excellent reviews [13,73,103,104] for further reading.

## 3. Interaction with Other Charged Proteins

Electrostatics plays a significant role in protein‒protein interactions [105,106]. Among the most famous examples, we can mention glyceraldehyde-3-phosphate dehydrogenase (GAPDH), which has positively charged groove and interacts with tubulin [107], alpha-synuclein [108], membrane transport proteins [109,110], and other acidic proteins. Interaction of serine proteases with inhibitors is also electrostatically driven [105]. Electrostatic interaction can also be important for protein recognition. Thus, translocation of negatively charged residues from the non-binding interface of membrane channel inhibitor BmP05 to the binding one resulted in switching of the binding site to a former non-binding interface [111,112]. The widely used barnase‒barstar system is also based on electrostatic interaction, which determines extremely high binding constant [113,114]. Electrostatic interactions can also play a role in an unfolded protein interaction with chaperones: for example, negatively charged GroEL interacts more efficiently with positively charged proteins [115]; the acidic region of small heat shock proteins is important for chaperone function [50].

How can MD simulations help in a study of electrostatically driven protein‒protein interactions? A “simple” (providing fewer quantitative atomistic details than MD) analysis of electrostatic potential can suggest potential binding sites [99,116,117]. For the complexes with a known structure, it can provide a rough estimation of the binding energy, and combined with point mutagenesis of charged residues, help to reveal the contribution of each residue to the binding [118,119]. However, in both cases, the MD simulations approach is the method of choice since it allows for an analysis of the optimized complex structure. Furthermore, quantitative analysis of MD simulations’ trajectories provides an accurate estimation of the binding energy, including energy decomposition [120,121,122]. In the aforementioned example of the barnase‒barstar complex, the analysis of rigid body complex is reasonable [123] but MD simulations provide much more detailed information about the formation of the complex, the contribution of different type of interactions, and the importance of each residue [124]. Since the structure of the barnase‒barstar complex is known, binding energy can also be estimated using steered MD simulations [125]. Then, MD simulations can provide insight into molecular mechanism of complicated processes if experimental investigation is hampered. This can be illustrated by computational studies of nascent protein release from ribosomal tunnel, which proposed the mechanism of the polypeptide chain movement and the role of charged residues of ribosomal proteins [126,127,128]. Finally (and maybe most importantly), MD simulation is a powerful approach for comparative investigation of similar proteins’ interaction and drug design. It is especially important if the binding is determined not only by electrostatic interactions. Thus, pro-apoptotic Bak peptide binds to anti-apoptotic proteins such as Bcl-2 through hydrophobic and electrostatic interactions [129], and the use of MD simulations allows for accurate comparison of the binding with different anti-apoptotic proteins and provides a basis for the detailed understanding of the mechanism of apoptosis regulation [130].

There are a lot of other examples of MD simulations’ utilization for modeling of charged protein behavior since electrostatic interactions are important for almost all systems. In the next two sections, we focus on two specific cases, where the impact of electrostatic interactions is very high: intrinsically disordered proteins and proteins undergoing post-translational modifications associated with charge changes.

## 4. Unfolded Proteins

MD simulations are a powerful approach to study unfolded proteins (especially intrinsically disordered proteins) and their complexes with other proteins [131] because of the difficulty of obtaining atomistic details from experimental methods. Furthermore, since intrinsically disordered proteins are known to have more charged residues (and therefore a higher net charge) in comparison with natively folded proteins [132], the modeling of electrostatic interactions is of special interest [133].

Thus, electrostatic interactions and the formation of intra-molecular and inter-molecular salt bridges is an important factor for amyloid aggregation [134]. This type of aggregation is associated with the formation of β-structured fibrils and is considered to be a reason for the development and progression of many human diseases [135]. Many amyloidogenic proteins are intrinsically disordered, and the mechanism of their amyloid conversion and aggregation is unclear. Numerous papers on MD simulations of amyloidogenic proteins at different levels (atomistic, coarse-grained, lattice simulations) are reviewed in [131,136,137,138,139]. Using MD simulations, electrostatics and formation of salt bridges were shown to influence the stability of the prion protein [140,141,142] as well as amyloid fibrils [143]. The role of familial mutations of α-synuclein charged residues was attributed to changes in fibril stability arising from alterations in electrostatic interactions [144]. A high impact of salt bridges was also shown for Aβ peptides, fibrils, and oligomer formation through β-turn stabilization [145,146,147]; the destabilization of intra-molecular salt bridges was suggested as a key factor in the copper ions’ binding effect on amyloid β fibrillization [148,149]. 

Formation of salt bridges is also important for the interaction of intrinsically disordered proteins with other proteins. Thus, α-synuclein and Aβ peptide are known to interact and form co-aggregates [150], and electrostatic interactions were suggested to be a key factor of the interaction [151,152]. The same was true for α-synuclein and β-synuclein co-oligomerization [153]. Electrostatically-driven binding was also shown for α-synuclein and GAPDH: the positively charged groove comprising substrate-binding site of GAPDH was predicted to be the binding site for α-synuclein, which was corroborated by experimental analysis of the α-synuclein effect on GAPDH enzymatic activity [108]. This interaction may be involved in Parkinson’s disease development via glycolysis efficiency decrease due to GAPDH inactivation [154]. 

To conclude, MD simulations can provide a detailed model of the interaction, aggregation, and sometimes folding of the unfolded proteins, which can then be experimentally verified. It is useful, for example, in comparative investigation of point mutations effect, which is important in the case of amyloidogenic peptides associated with neurodegenerative diseases [144,147]. Notably, modeling of the unfolded protein (or flexible peptide) interactions requires multiple simulations, replica-exchange MD simulations, umbrella sampling, Markov state models, or other techniques to obtain reliable information about the conformational ensemble [133]. In addition, the “usual” force fields do not describe unfolded proteins’ behavior properly, so force fields designed specifically for unfolded proteins should be used: for example, full-atom force fields CHARMM36m [155], ff14IDPSFF [156], and a99SB-disp [157].

## 5. Effect of Post-Translational Modifications

### 5.1. Phosphorylation

Phosphorylation is one of the most common protein post-translational modification, and it is associated with charge change [158]. Though hundreds of papers describe the mechanism of phosphorylation-mediated regulation of protein behavior and protein‒protein interactions on the basis of experimental data [159,160,161], MD simulations often help to reveal atomistic details of the mechanism [162,163]. Since the number of such papers is very large (almost 700 papers are indexed in PubMed with the keywords “MD simulation” and “protein phosphorylation”), we discuss only some examples of successful usage of MD to study the role of phosphorylation.

The related papers can be separated into two groups according to the mechanism of the phosphorylation contribution revealed by MD simulations. The first one is *direct modulation of protein‒protein interaction* via changes in local electrostatics and formation or disruption of new inter-molecular contacts. Phosphorylation can lead to a significant change in the protein surface’s properties [164] and therefore in its behavior. Thus, electrostatic repulsion between phosphorylated site of transcription factor p53 and a negatively charged patch of ubiquitin ligase MDM2 (mouse double minute 2 homolog) was suggested to be a reason for the complex dissociation, which occurs due to p53 phosphorylation [165]. The same can be true in the case of other charged biological macromolecules. For example, phosphorylation of the transmembrane peptide phospholamban is known to regulate the activity of the sarcoplasmic reticulum calcium pump in cardiac muscle. MD simulations suggested that phosphorylation enhances the interaction of the cytoplasmic part of phospholamban with the phospholipid bilayer via the formation of contacts between the protein phosphate group and a lipid ammonium group, thus inhibiting its interaction with other calcium pump proteins and inhibiting the action of the pump [166]. However, another work suggested a different mechanism: formation of new intra-molecular bonds between phosphate group and positively charged residues stabilizes specific conformation of the phospholamban [167]. This second mechanism, i.e., *changes in protein conformation induced by the formation of new intra-molecular contacts* (usually salt bridges), was proposed to be a key factor in many cases and seems to be much more widespread. Thus, electrostatic interaction between phosphorylated serine and arginine located in different domains destabilizes the closed conformation of DNA polymerase β and consequently inhibits its activity, as was suggested using MD simulations and corroborated with experiments [168]. On the contrary, phosphorylation was shown to lock active conformation of c-Src kinase, thus providing allosteric regulation of the enzyme functionality [169]. Phosphorylation can stabilize α-helices by the formation of additional intra-helical bonds with neighboring positively charged arginine or lysine [170]. Phosphorylation of the Shc adaptor protein changes its flexibility and thus indirectly influences the interaction with the receptor without direct interaction of the phosphorylated tyrosine with the receptor protein [171]. Phosphorylation is also important for the behavior of unfolded proteins, including those associated with neurodegenerative diseases [172]. According to the results of modeling, the role of phosphorylation can arise from the formation of new intra-molecular contacts that stabilize the secondary or tertiary structure of the protein, as was shown for tau peptide [173] and initiation factor 4E-binding protein 2 [174].

In conclusion, phosphorylation usually leads to “cosmetic” changes such as the formation of a salt bridge, which is nevertheless important for the protein behavior. It suggests a high capability of the MD simulations to provide insights into the mechanism of such signal transduction from the local protein area to the overall structure and function. The main conclusions (but usually not atomistic details) obtained using MD simulations can and should be experimentally verified.

### 5.2. Sulfation

The appearance of a charged group is of special importance for sulfated proteins. Protein sulfation usually occurs at tyrosine [175] and sometimes at threonine or serine [176] residues. It is a relatively rare (compared to phosphorylation) post-translational modification: sulfation and phosphorylation were confirmed experimentally for fewer than 200 and more than 15,000 proteins, respectively [177]. Typically, the sulfation site is located in the acidic area of the protein surface, and the insertion of the sulfate group leads to an increase of the negative charge [178]. As a result, sulfation alters protein‒protein interaction. Thus, sulfation of some neuropeptides and toxins activates or inhibits their activity [178,179]; the sulfation of proteins involved in blood coagulation systems significantly enhances the respective binding constants [4,5].

Unfortunately, the atomistic details of the role of sulfation in the behavior of the aforementioned proteins and peptides are scant. Based on our data about protein affinity to sulfate- and phosphate-based polymers, we hypothesized that sulfation occurs when a strong interaction is required, whilst phosphorylation is preferable for less tight but reversible interactions [177]. This hypothesis agrees with the data on the MD simulations of tyrosine-sulfated V2 peptide corroborated with experiments [180]. Two more papers [181,182] directly compare the behavior of sulfated and unmodified proteins as well as sulfated and phosphorylated peptides, but the conclusions seem to be questionable because of unclear analysis and interpretation. Finally, we should mention the second hypothesis on the role of protein sulfation, suggested using the MD simulations approach: facilitating intra-molecular contacts by sulfation restricts the flexibility of the protein region. This hypothesis was corroborated experimentally for a particular case: the difference in flexibility was proven by NMR spectroscopy, and interaction of sulfate group with the receptor protein was not observed in the crystal structure of the complex [183].

Summarizing this scarce information, we suggest MD simulations as a powerful tool to understand the role of sulfation in particular cases of protein‒protein interaction. We wonder why the use of this approach is so rare and expect that the growing usage of MD simulations will provide insight into the role of protein sulfation in other numerous cases.

### 5.3. Glycation

Glycation of positively charged residues (lysine and arginine) is a common non-enzymatic process occurring in living cells [184] as well as under artificial conditions such as food processing and cooking [185]. Glycation of proteins involved in the development and the progression of human diseases—diabetes mellitus and neurodegenerative diseases—is of special interest [184,186,187]. Various end-products appearing due to glycation by sugars and aldehydes (glucose, methylglyoxal, glyceraldehyde-3-phosphate, etc.) change the protein charge and can influence protein functionality and interaction with other biological macromolecules [188,189]. The progression of neurodegenerative diseases such as Alzheimer’s and Parkinson’s diseases in diabetes seems to be associated with these processes [154,188,190,191].

Modeling of glycated proteins is very complicated because of a large diversity of probable intermediate and end products (so-called advanced glycation end products) for each residue and almost statistical distribution of the glycation sites among lysine and arginine residues [6,7]. Furthermore, some of them involve intra-molecular or inter-molecular cross-linking. On the other hand, the same reasons together with the difficulty of the controlled preparation of the particular glycation end product in vitro make experimental investigation of glycated proteins very complex and suggest molecular modeling of probable glycation products as the method of choice. Firstly, MD simulations can be used together with docking (or solely) to predict the binding site for the glycation agent or compare the binding of different glycation agents and to analyze conformational changes of the protein structure [192,193,194,195]. Then, MD simulations can also be used to analyze conformational changes caused by glycation, i.e., formation of the advanced glycation end product [196,197,198,199,200,201]. Generally, glycation of positively charged residues (lysine and arginine) results in the formation of negatively charged (carboxymethyl lysine or carboxyethyl lysine), neutral (pyrraline), or cross-linked group with the +1 charge instead of two positively charged residues (pentosidine, glyoxal lysine dimer) [6,7,187,191]. Therefore, the analysis of electrostatic interaction changes is of special importance for understanding the effect of glycation on the protein structure and behavior. For example, the ligand/substrate binding can be influenced by the disappearance of the salt bridge between the ligand and the modified arginine [202] or local changes in electrostatic potential in the case of a negatively charged ligand [203], as well as glycation-driven conformational changes of the protein structure [197,204]. Finally, simulation of the intra-molecular [197] and inter-molecular [205] cross-linked advanced glycation end products is also possible, but prediction of the preferred cross-linking sites is a key point of such modeling. In case of a lack of experimental information, the lysine glycation prediction server NetGlycate can be used [206]. As an alternative, the most biologically important residues can be considered as glycation sites. Notably, the last approach suggests a hypothesis for how glycation *might influence* protein function, but cannot prove how it *actually does influence* it.

Intrinsically disordered amyloidogenic proteins are a beautiful example of the glycation effect on charge-based interactions revealed using MD simulations. For example, according to the results of modeling of Aβ peptide self-oligomerization [207], glycation (carboxymethyl lysine modification was used) resulted in an increase of the beta sheet content and formation of stronger oligomers due to an enhanced salt bridging between the monomers as compared to non-modified peptide. Furthermore, electrostatic interactions were also attributed to an increased stability of glycated pre-formed protofibrils. The enhanced amyloidogenic aggregation level was corroborated experimentally. The second example is an effect of glycation of α-synuclein on its interaction with GAPDH, which was significantly enhanced by glycation [208]. Substitution of positively charged lysine residues with negatively charged carboxymethyl lysine in the N-terminal region induced the binding by almost entire α-synuclein molecule, while non-modified α-synuclein interacted with the GAPDH anion-binding groove only by the negatively charged C-terminal region (Figure 3). The change in the α-synuclein charge, i.e., the applicability of the used model, as well as the increase in the protein‒protein interaction efficiency, was proven by experimental approaches. Notably, only indirect experimental examination of the overall effect of glycation was performed in both discussed examples, whereas the specific glycation end products were not determined.

### 5.4. Cysteine Oxidation

Thiol groups of cysteine can be subjects of oxidation up to formation of charged groups such as sulfenic and sulfonic acids [209,210,211]. In most cases, such non-enzymatic modification has one of two outcomes: disulfide bond breakage or inactivation of a key catalytic residue that strongly affects the protein functionality [212]. However, deep oxidation of cysteine residues can also influence protein interaction with other biological macromolecules, and this effect can arise from charge changes [213,214]. According to computational studies, cysteine oxidation may result in the formation of new H-bonds as well as new electrostatic interactions [215], and influence protein structure [216,217]. Unfortunately, information about MD simulation of such systems is very scarce, probably because of the diversity and high reactivity of intermediate products, which requires simulation of different modifications and comprehensive usage of QM/MM approach [218], especially if the modified cysteine is located in the active site of the enzyme. 

## 6. Concluding Remarks

Summarizing the reviewed data, we conclude that electrostatic interactions are critically important for protein behavior since they control interaction with other protein, glycosaminoglycans, polyphosphates, and nucleic acids. The molecular dynamics simulations approach provides an outstanding opportunity to probe the atomistic details of these interactions. Furthermore, it is the method of choice for comparative study of the role of point mutations and post-translational modifications, as well as for drug design. It is of special interest for intrinsically disordered protein, including amyloidogenic proteins and peptides, since many amyloidosis-related diseases are associated with point mutations. Notably, electrostatic interactions and the formation of intra- or inter-molecular salt bridges are important to the behavior of amyloidogenic proteins. In addition, these unfolded proteins do not have a rigid structure involved in molecular recognition. Hence, the modeling of electrostatic interactions helps to elucidate the mechanism of their interaction with other macromolecules present in cells. There are still challenges to modeling the aforementioned systems, including proper sampling and development of realistic force fields. Finally, many important post-translational modifications associated with changes in the protein local charge are difficult to study using experimental approaches, and molecular dynamics simulations of such systems will elucidate the role of these modifications on protein‒protein interactions. In the case of phosphorylation, molecular modeling can be considered a routine approach, while in the case of sulfation, glycation, or polyphosphorylation, it should provide insight into the molecular mechanism of the unclear impact of the modification. We expect that the number of papers employing molecular dynamics simulations for post-translational modifications investigation will increase. Specifically, this approach should force an understanding of the role of intrinsically disordered proteins post-translational modifications in human pathologies such as diabetes, aging, neurodegenerative diseases, etc. There is a large body of evidence for the interconnection of the listed diseases’ development and progression and protein post-translational modifications, but the mechanism underlying this relation is unclear. Notably, this combination, i.e., unfolded proteins and non-enzymatic post-translational modifications, is the most complicated for in vivo study, and therefore the capabilities of molecular modeling would be outstandingly useful.

## Figures and Tables

**Figure 1 ijms-20-01252-f001:**
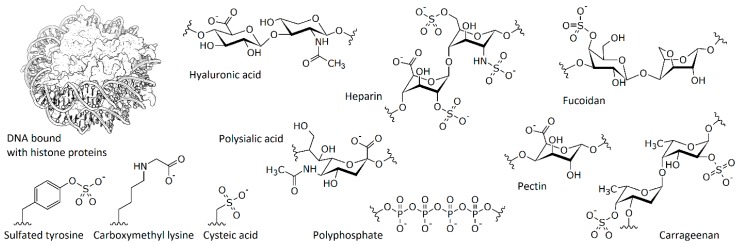
The most important naturally occurred charged polymers and examples of charge-associated post-translational modifications.

**Figure 2 ijms-20-01252-f002:**
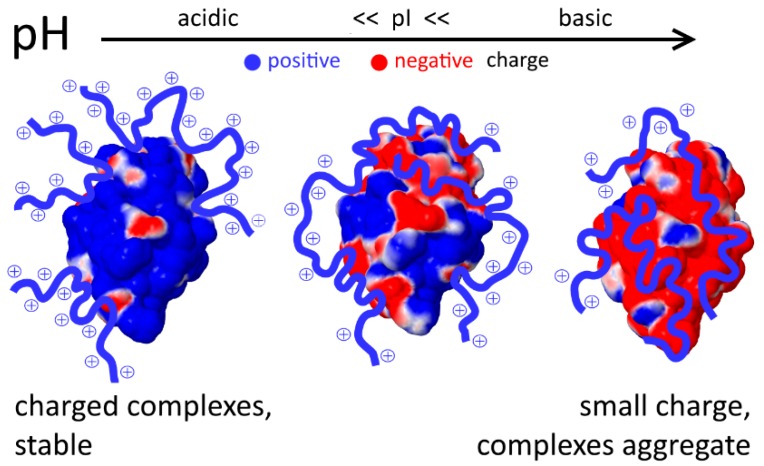
Chaperone-like activity of charged polymers. Adopted from [67].

**Figure 3 ijms-20-01252-f003:**
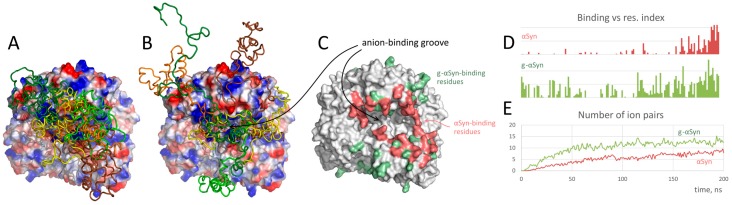
MD simulations of GAPDH binding with α-synuclein: five typical position of intact (**A**) and glycated (**B**) α-synuclein shown in cartoon on GAPDH surface colored according to electrostatics; binding residues (**C**); binding profiles on α-synuclein sequence (**D**); number of ion pairs formed between GAPDH and different forms of α-synuclein (**E**) [208]. Red and green colors in **C**–**E** represent the data for native and glycated forms of α-synuclein, respectively.

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
