# Peer review of "Protein Interaction with Charged Macromolecules: From Model Polymers to Unfolded Proteins and Post-Translational Modifications"

_ijms, 2019, doi:10.3390/ijms20051252_

Reviewer 1 Report

In this review, the authors addressed the progress in studies of protein interaction with charged macromolecules, such as DNA and RNA, polyphosphates, sulfated glycosaminoglycans and charged proteins, mainly focusing on the technique molecular dynamic (MD) simulations. The manuscript focuses on an interesting subject in the field. I would recommend the manuscript to publish in Int. J. Mol. Sci. once the following comments below are addressed.

(1)   The review is focused on MD simulations studies on the electrostatic interactions of proteins with other molecules. How the electrostatic interaction is expressed should be described. The advantage and limitation of this technique, as well as the future perspective on this subject should be addressed.

(2)   The organization of the manuscript is elusive, such as the section “3. Counter ions release effect” seems not parallel with other section. Another example is that in the third paragraph in page 3, a general feature of charged polymers interacting with protein surface was addressed, therefore the paragraph arrangement makes the manuscript obscured.

(3)   Although the authors mentioned “we comprehensively compare the data from molecular dynamics simulations with experimental data throughout the review”, actually, how the simulation results were corroborated by experiments was not well addressed in the manuscript.

(4)   Some sentences in the manuscript need English proof-reading, such as line 77-79 in page 2, line 273-274 in page 7.

Author Response

In this review, the authors addressed the progress in studies of protein interaction with charged macromolecules, such as DNA and RNA, polyphosphates, sulfated glycosaminoglycans and charged proteins, mainly focusing on the technique molecular dynamic (MD) simulations. The manuscript focuses on an interesting subject in the field. I would recommend the manuscript to publish in Int. J. Mol. Sci. once the following comments below are addressed.

We thank the Reviewer for the consideration of our manuscript and many critical comments, which should improve our manuscript. The changes are colored in yellow.

(1)   The review is focused on MD simulations studies on the electrostatic interactions of proteins with other molecules. How the electrostatic interaction is expressed should be described. The advantage and limitation of this technique, as well as the future perspective on this subject should be addressed.

We have added a short description of the electrostatic interactions and discussed the challenges important for realistic modeling of their contribution (l.125-131 and the next paragraph, l.256-259, l.330-331, l.366-367).  We also have added some suggestions on future perspectives into the Concluding remarks (l.415-430).

(2)   The organization of the manuscript is elusive, such as the section “3. Counter ions release effect” seems not parallel with other section. Another example is that in the third paragraph in page 3, a general feature of charged polymers interacting with protein surface was addressed, therefore the paragraph arrangement makes the manuscript obscured.

The section 3 was aimed to summarize the “technical” lessons learned from modeling of protein interactions with model polymers. Since some of lessons were mentioned in the section 1, we have reconstructed this part of the manuscript according to the comments 1-2. First, we combined sections 2 and 3 since most information in section 3 was about model polymers, and rearranged the paragraphs (l.125-180, main part of the previous Section 3 is colored in gray). We believe the paragraphs sequence became more logical.

(3)   Although the authors mentioned “we comprehensively compare the data from molecular dynamics simulations with experimental data throughout the review”, actually, how the simulation results were corroborated by experiments was not well addressed in the manuscript.

We have added the discussion of the relation between MD results and experimental data in few cases where it was missed (l.99-100, l.299-303, l.324-327, l.374-375, l.379-383).

(4)   Some sentences in the manuscript need English proof-reading, such as line 77-79 in page 2, line 273-274 in page 7.

We have revised aforementioned examples and performed proofreading of the manuscript (not marked).

Reviewer 2 Report

In this manuscript, the authors tried to review recent simulations study concerning protein interaction with charged macromolecules. Unfortunately, they failed to give a comprehensive review on the designated topics. It has been clearly stated in the Journal scope that a review should be made in a wide range, not in a mini way. Thus the current form of this paper is not suitable for the journal. Moreover, the content of the manuscript is quite incomplete. The Introduction is too short. The modeling section gives almost no information about how to model the proteins and nucleic acids in simulations. It simply re-directed the readers to read the papers referred. The manuscript then jumps quite arbitrarily to review some topics such as counterion release, charged proteins, unfolded proteins, and post-translational modifications. These subjects are piecewise and no connections have been made between them in the text.The logic flow is not clear at all. The presentation is bad. Through reading the manuscript, readers can hardly obtain any useful information from the papers reviewed. The value of this manuscript is therefore low. I do not recommend it for publication.

Author Response

In this manuscript, the authors tried to review recent simulations study concerning protein interaction with charged macromolecules. Unfortunately, they failed to give a comprehensive review on the designated topics. It has been clearly stated in the Journal scope that a review should be made in a wide range, not in a mini way. Thus the current form of this paper is not suitable for the journal. Moreover, the content of the manuscript is quite incomplete. The Introduction is too short. The modeling section gives almost no information about how to model the proteins and nucleic acids in simulations. It simply re-directed the readers to read the papers referred. The manuscript then jumps quite arbitrarily to review some topics such as counterion release, charged proteins, unfolded proteins, and post-translational modifications. These subjects are piecewise and no connections have been made between them in the text. The logic flow is not clear at all. The presentation is bad. Through reading the manuscript, readers can hardly obtain any useful information from the papers reviewed. The value of this manuscript is therefore low. I do not recommend it for publication.

We thank the Reviewer for a critical review of our manuscript. We have made many changes according to the comments. We have expanded the Introduction. We added some details about general principles of MD simulation of nucleic acids and other charged polymers. However, we suppose that repeating of information about modeling of protein-nucleic acids complexes in the current manuscript would be meaningless since there are many excellent comprehensive reviews on this topic. We have cited them and just focused on the role of charged moieties. Then, we rearranged paragraphs and removed section 3 to make the logic flow more clear. Connections between the sections have been added. Starting from the model polymers, we analyzed the lessons learned from such model systems, and then switched to protein-protein interactions. Some specific cases – unfolded proteins and proteins undergoing charge-associated post-translational modifications – were considered in more details because of significant role of electrostatic interactions in such systems. The changes are marked with yellow. We also performed proofreading of the manuscript (not marked).

Reviewer 3 Report

The manuscript by Semenyuk and Muronetz addresses an important and interesting issue, the interaction between charged macromolecules and proteins, with a focus on MD-based methods. The authors provide a good overview on this topic and discuss the relevant literature in sufficient detail. I can therefore recommend publication in IJMS.

Minor points:

1) Due to the broad readership of IJMS, a brief introduction into MD would be desirable.

2) Ribosomes represent another relevant example, but this topic is missing in the review article.

Author Response

The manuscript by Semenyuk and Muronetz addresses an important and interesting issue, the interaction between charged macromolecules and proteins, with a focus on MD-based methods. The authors provide a good overview on this topic and discuss the relevant literature in sufficient detail. I can therefore recommend publication in IJMS.

We thank the Reviewer for the consideration of our manuscript and appreciate the positive comments. We have made changes according to the following comments.

Minor points:

1) Due to the broad readership of IJMS, a brief introduction into MD would be desirable.

We have added a short description into the section 1. Introduction (l.52-59).

2) Ribosomes represent another relevant example, but this topic is missing in the review article.

Thank you for the suggestion. We have included this example (l.41, l.205-209).

Round  2

Reviewer 2 Report

The authors have made efforts to improve their manuscript. The logic flow and the presentation is now better. The reviewed topics may interest some people working in the domain concerning protein interaction with charged molecules. I’ll recommend the paper for publication if the authors could fix the following problem.

The abbreviations given at Line 434 are not complete at all. There are still so many in the paper, for example, QM/MM, GLYCAM, ATP, ADP, etc. Please fix it.

Author Response

The authors have made efforts to improve their manuscript. The logic flow and the presentation is now better. The reviewed topics may interest some people working in the domain concerning protein interaction with charged molecules. I’ll recommend the paper for publication if the authors could fix the following problem.

We thank the Reviewer for the consideration of our manuscript and appreciate the positive mark of the changes.

The abbreviations given at Line 434 are not complete at all. There are still so many in the paper, for example, QM/MM, GLYCAM, ATP, ADP, etc. Please fix it.

We have added the missed abbreviations (l.434). In addition, we described abbreviation PAMAM but did not included it in the Abbreviation list since it was used only once (l.105). GLYCAM is a name of the force field, which contains parameters for numerous carbohydrates. We have added the corresponding comment (l.86).